# Surfactant-Modified CdS/CdCO_3_ Composite Photocatalyst Morphology Enhances Visible-Light-Driven Cr(VI) Reduction Performance

**DOI:** 10.3390/nano12213923

**Published:** 2022-11-07

**Authors:** Wen-Yi Wang, Tian Sang, Yan Zhong, Chao-Hao Hu, Dian-Hui Wang, Jun-Chen Ye, Ni-Ni Wei, Hao Liu

**Affiliations:** 1School of Materials Science and Engineering, Guilin University of Electronic Technology, Guilin 541004, China; 2Guangxi Key Laboratory of Calcium Carbonate Resources Comprehensive Utilization, Hezhou University, Hezhou 542899, China; 3Guangxi Key Laboratory of Information Materials, Guilin University of Electronic Technology, Guilin 541004, China

**Keywords:** surfactant, CdS/CdCO_3_, photocatalytic, Cr(VI)

## Abstract

The surfactant modification of catalyst morphology is considered as an effective method to improve photocatalytic performance. In this work, the visible-light-driven composite photocatalyst was obtained by growing CdS nanoparticles in the cubic crystal structure of CdCO_3_, which, after surfactant modification, led to the formation of CdCO_3_ elliptical spheres. This reasonable composite-structure-modification design effectively increased the specific surface area, fully exposing the catalytic-activity check point. Cd^2+^ from CdCO_3_ can enter the CdS crystal structure to generate lattice distortion and form hole traps, which productively promoted the separation and transfer of CdS photogenerated electron-hole pairs. The prepared 5-CdS/CdCO_3_@SDS exhibited excellent Cr(VI) photocatalytic activity with a reduction efficiency of 86.9% within 30 min, and the reduction rate was 0.0675 min^−1^, which was 15.57 and 14.46 times that of CdS and CdCO_3_, respectively. Finally, the main active substances during the reduction process, the photogenerated charge transfer pathways related to heterojunctions and the catalytic mechanism were proposed and analyzed.

## 1. Introduction

Environmental pollution caused by population expansion and industrial production is becoming increasingly prominent, resulting in serious damage to natural water environments [1,2,3,4,5]. In particular, toxic-heavy-metal ions, such as hexavalent chromium (Cr(VI)), are serious threats to human health [6]. It is pressing to develop long-life and efficient environmental purification technologies. Photocatalytic technology has become the focus in the field of environmental governance due to its low energy consumption, high stability, and sustainability [7,8,9]. In recent reports, the appropriate band gap of the material and the separation efficiency of electron-hole pairs were the main factors affecting the catalytic performance [10,11]. Based on previous studies, we found that CdS has a suitable band gap, and can reduce hexavalent chromium ions [12,13], which were reported as promising photocatalysts. Yu prepared CdS nanorods by using the solvothermal method [14]; Zhukovskyi successfully obtained CdS nanosheets with uniform thickness and controllable size by thermal decomposition [15]; and Liu used the solvothermal method to prepare CdS nanowires with uniform diameters [16].

In addition, the photocatalytic efficiency of the catalyst also largely depends upon the morphology, grain size and specific surface area of the material [17]. Dharamalingam et al. found that adding an appropriate amount of sodium dodecyl sulfate during the synthesis of MoS_2_ nanosheets could change the crystal orientation and structural morphology of bulk MoS_2_ [18]; Fang used a surfactant-assisted hydrothermal method to prepare flower-like ZnTi-LDHs catalysts with excellent methyl-orange-degradation performance [19]; Yang regulated micro spherical CdS into dendrites through sodium dodecyl sulfate [20]; and Flores studied the effects of various surfactants on the morphology of perovskite-type TiO_2_, which proved that TiO_2_ had the largest specific surface area and the best photocatalytic activity in the regulation of sodium dodecyl sulfate [21]. Therefore, it is important to study the relationship between catalyst activity and surface morphology.

In recent years, CdCO_3_ has been increasingly used in the field of photocatalysis as a wide-band-gap material. Zhang innovatively synthesized SnO_2_/CdCO_3_/CdS ternary heterojunction and achieved the efficient removal of U(VI) under visible light [22]; a CdCO_3_/RP (red phosphorus) composite photocatalyst was prepared by Xuan, which significantly improved the photocatalytic performance of RP [23]; Vidyasagar successfully prepared organic hybrid inorganic nano photocatalyst g-C_3_N_4_/CdCO_3_ by using the in situ microwave heating method, and achieved the efficient degradation of indigo carmine and the inactivation of Gram-negative Escherichia coli [24]. However, few studies have reported the changes in surfactants in the morphology of CdCO_3_ or the effect of its catalytic performance. In this study, CdS/CdCO_3_ composite catalysts were prepared by using the chemical-precipitation method and the relationship between the change in CdCO_3_ morphology and photocatalytic activity using an appropriate amount of sodium dodecyl sulfate was studied. The reduction effect of the catalyst on Cr(VI) ions and its photocatalytic reduction kinetics were investigated. The main active substances in the photocatalytic process were identified by trapping-agent experiments, and a catalytic path was proposed. The purposes of this study were to provide ideas for insulator morphology modification and to provide a reference scheme for semiconductor/insulator composite photocatalyst in the field of sewage purification.

## 2. Experimental Section

### 2.1. Materials

Cadmium sulfide (CdS) was procured from MACKLIN (China). Diphenylcarbazide (C_13_H_14_N_4_O) and cadmium chloride hemi (pentahydrate) (CdCl_2_·2.5H_2_O) were purchased from Aladdin (China). Acetone (CH_3_COCH_3_), sodium dodecyl sulfate (C_12_H_25_SO_4_Na), sodium carbonate (NaCO_3_), potassium dichromate (K_2_Cr_2_O_7_), nitric acid (HNO_3_), isopropyl alcohol (C_3_H_8_O), 1,4-benzoquinone (C_6_H_4_O_2_) and ethylenediaminetetraacetic acid disodium salt (C_10_H_14_N_2_Na_2_O_8_) were procured from XILONG SCIENTIFIC (China).

### 2.2. Preparation of CdS/CdCO_3_ Nanocomposites

A total of 0.4567 g of CdCl_2_·2.5H_2_O was dissolved in 50 mL of deionized water and the solution was subjected to 30 min stirring. The obtained solution was denoted as solution A. A total of 0.2119 g Na_2_CO_3_ was dissolved in 50 mL deionized water and stirred for 30 min, after which 1.4446 g CdS were added under continuous stirring until fully mixed. This solution was denoted as solution B. Solution A was slowly added into solution B, stirred for 12 h and filtered with a suction filter, after which it was dried in an oven at 60 °C for 12 h and ground. According to the different molar ratios of CdS and CdCO_3_, the composite catalyst was named *X*-CdS/CdCO_3_ (*X* = 6, 5, 2 and 1), where *X* represents the molar ratio.

### 2.3. Preparation of CdS/CdCO_3_@SDS Nanocomposites

In the process of synthesizing 5-CdS/CdCO_3_, an appropriate amount of sodium dodecyl sulfate (SDS) was added when solution A and solution B were mixed and, after stirring for 12 h, the remaining steps were consistent with the above experiments.

### 2.4. Characterizations

A powder X-ray diffraction (SmartLab 9kw XRD, Tokyo, Japan) apparatus with 2*θ* range from 10° to 80° was used to analyze the phase and crystal structure of the samples. More detailed structural examinations of the samples were determined using scanning-electron microscopy (JEOL JSM-7610F SEM, Tokyo, Japan) and transmission-electron microscope (Tecnai G2 F30 S-TWIN TEM, Waltham, MA, USA). The specific surface areas of photocatalysts were analyzed by a Micromeritics ASAP 2020 HD88 (BET, St. Louis, MO, USA). X-ray photoelectron spectra (XPS, USA) were obtained by Thermo Fisher ESCALB 250xi with Al Ka radiation and the peaks were calibrated by C 1 s at 284.6 eV. The catalytic performances of the photocatalysts were investigated using a UV-vis spectrophotometer (Shimadzu, UV-2600, Kyoto, Japan) with 300 W xenon light (CEAULIOHT CEL-HXF300, Beijing, China) source and a 420-nanometer cut-off filter. The electrochemical measurements were performed in a conventional three-electrode cell on a CS310H electrochemical workstation.

### 2.5. Photocatalytic Reduction Studies of Cr(VI)

The photocatalytic activity of the CdS, CdCO_3_ and *X*-CdS/CdCO_3_ were analyzed by assessing the visible-light-induced reduction of Cr(VI) in aqueous solution. For photoreduction reaction, 5 mL of hexavalent chromium ion solution with a concentration of 10 mg/L were diluted with 95 mL water. Subsequently, 0.2 g C_13_H_14_N_4_O were dissolved in a mixture of 50 mL CH_3_COCH_3_ and 50 mL deionized water. Finally, the two solutions were mixed evenly. Photocatalytic experiments were carried out for 30 min and, before the experiment started, 50 mg of photocatalyst were added to the target solution alone. The mixture was stirred for 30 min in a dark environment to reach adsorption–desorption equilibrium before the photoreaction. Next, the solution was irradiated with stirring under visible light. Every 10 min, 3 mL of sample were analyzed using a UV-Vis absorption spectrophotometer.

The Cr(VI) photoreduction efficiency was calculated by following Equation [25].
(1)$$\mathrm{Photoreduction}\;\mathrm{rate}=\frac{C_{0}-C_{t}}{C_{0}}\;\times \;100\%$$
where $C_{0}$ is the initial dye concentration (mg/L) and $C_{t}$ is the concentration (mg/L) of dye after a certain irradiation time *t.*

The photocatalytic process followed Equation [26].
(2)$$\ln C_{0}/C_{t}=kt$$

Herein, *k* is the apparent pseudo-first-order rate constant.

## 3. Results and Discussion

### 3.1. XRD and XPS Analysis

The X-ray diffraction (XRD) measurements of the power samples and the XPS spectra were performed to identify the crystal structure and chemical composition of CdS, CdCO_3,_ *X*-CdS/CdCO_3_ and 5-CdS/CdCO_3_@SDS, respectively. Figure 1a shows the XRD data of the synthesized monophase and composites. Apparently, the main characteristic peaks of all the composites were indexed to CdS (JCPDS 10-0454, *F*-43*m*) and CdCO_3_ (JCPDS 42-1342, *R*-3*c*), which revealed that all the composite photocatalysts were successfully synthesized without other impurity phases. Notably, the corresponding diffraction peaks in the composite photocatalysts became stronger when the content of the CdCO_3_ increased gradually, and the position of the peak did not change. The intensity of the diffraction peaks of the 5-CdS/CdCO_3_@SDS was weakened, which may have been caused by the refinement of the grain size of the composite catalyst under the action of the surfactants [27]. The elemental composition and valence state of the as-prepared samples were measured by XPS. Figure 1b exhibits the survey spectra of the 5-CdS/CdCO_3_@SDS. The picture demonstrates the existence of C, Cd, O and S elements without any other impure elements. The XPS spectra of the C element (Figure 1c) shows that the characteristic peaks of C 1 were located at 284.8 eV and 289.7 eV, which belonged to the calibration and O–C=O bonds [28], respectively. The Cd 3d spectrum (Figure 1d) can be dissociated into two peaks; the binding energies at 405.1 eV and 411.9 eV were attributed to Cd 3d_5/2_ and Cd 3d_3/2_, which confirmed that the chemical state of the Cd element was Cd^2+^ in the cadmium carbonate. The peak of the O element at 531.7 eV suggested that the chemical state of the O element was O^2−^ in the CdCO_3_ [29]. Meanwhile, the peaks located at 161.4 eV and 162.6 eV corresponded to S 2p_3/2_ and S 2p_1/2_ of the CdS.

### 3.2. SEM, TEM and BET Analyses

Figure 2 displays the SEM images of CdS, CdCO_3_, 5-CdS/CdCO_3_ and 5-CdS/CdCO_3_@SDS. As shown in Figure 2a,b, CdS exhibited agglomerated microspheres with diameters of about 100–200 nm. For the CdCO_3_ sample, a 3D-cube block composed of varying of numbers nanosheets with diameters of 300–400 nm is shown in Figure 2c,d. The as-prepared 5-CdS/CdCO_3_ maintained the 3D-cube block. The small CdS nanospheres were grown uniformly on the surface of the flake CdCO_3_ after precipitation treatment (Figure 2e,f) and confirmed the successful synthesis of the 5-CdS/CdCO_3_ composite. The morphology of the composite catalyst was significantly changed after the modification of the surfactants (Figure 2g,h). We can clearly see that the cube structures of the CdCO_3_ composed of nanosheets were transformed into porous ellipsoid spheres composed of nanoparticles and that unshaped CdS nanoparticles were tightly attached to the surface of CdCO_3_ nanospheres. As a hydrophilic anionic surfactant, sodium dodecyl sulfate (SDS) had a steric hindrance effect and reduced the surface tension [30,31]. After adding SDS during the formation of CdCO_3_, the oxygen atoms in the hydrophilic group of SDS coordinated with the atoms on the surface of the CdCO_3_, leaving the long chain of hydrophobic groups of C-C alkyl to stretch around the CdCO_3_, surrounding its surface and reducing its surface energy. The unmodified CdCO_3_ particles were composed of nanosheets of different shapes and the system’s surface energy was very high [17]. Therefore, from a thermodynamic point of view, originally protruding surfaces self-assemble through hydrogen bonds and tend to clump together to attenuate the surface energy, eventually forming elliptical-like nanospheres [32].

To study the structure of the 5-CdS/CdCO_3_@SDS heterojunction, the composites were analyzed by transmission-electron microscope (TEM). The TEM image (Figure 3a) revealed the presence of nano-ellipsoids ~200 nm wide and ~400 nm long, whose surfaces were decorated with many CdS nanoparticles, which also proved that there was some kind of interaction between the two materials (Figure 3b). The HR-TEM image (Figure 3c) showed obvious lattice spacing of 0.215 nm and 0.166 nm, which was consistent with the (200) crystal plane of the CdS and the (012) crystal plane of the CdCO_3_. These two crystal faces corresponded to one of the strongest diffraction peaks of the two single phases, respectively. The mapping-elements analysis confirmed the uniform distribution of the Cd, C, O and S, further proving the simultaneous presence of cadmium carbonate and cadmium sulfide. The specific surface areas of the four samples are exhibited in Table 1. The experimental results showed that the addition of surfactants improved the specific surface area of the composite catalyst and confirmed the refinement of the grains, which was consistent with the weakening of the XRD diffraction peak of the sample of 5-CdS/CdCO_3_.

### 3.3. Photocatalytic Reduction and Influence Factor

The experiments on the photoreduction of Cr(VI) were conducted to assess the photocatalytic activity of all the as-prepared samples. As shown in Figure 4a, the CdS and CdCO_3_ exhibited poor photocatalytic activity, while the photocatalytic performances of the composite catalysts were significantly enhanced compared with the performances of the two single-phase catalysts. In particular, the reduction efficiency of the sample 5-CdS/CdCO_3_@SDS was increased by about 10% compared with no SDS added, which may have been due to the increase in the specific surface area of the composite catalyst after the modification of the surfactant and the fuller contact with the Cr(VI) solution. In Figure 4b, the reduction-rate constant of the Cr(VI) can be clearly seen. The reduction rate of the 5-CdS/CdCO_3_@SDS was 0.0675 min^−1^, which was 15.57 times and 14.46 times that of the CdS and CdCO_3_, respectively. In Figure 4c, the absorption-peak intensity of the Cr(VI) at 540 nm gradually decreases with the increasing of irradiation time. The capture-agent experiment was used to further study the active substances in the photocatalytic process. The Edta-2Na, BQ and IPA captured the h^+^, •O_2_^−^ and •OH, respectively. Furthermore, it can be seen from Figure 4d that BQ and IPA showed obvious inhibitory effects on the photocatalytic process, indicating that the main active substances in the photoreaction process were •O_2_^−^ and •OH. At the same time, the sample showed good cycle performance (Figure 4e), and the photocatalytic reduction efficiency reached 72.4% after three rounds (1.5 h) of cycle tests.

To verify the enhanced interfacial separation efficiencies of the photogenerated electron-hole pairs of 5-CdS/CdCO_3_@SDS, the transient photocurrent responses of the catalysts were tested under visible light for five light on –off cycles (Figure 5a). During transient photocurrent measurement, an Ag/AgCl electrode was used as the reference electrode and a Pt foil electrode acted as the counter electrode. The working electrodes were designed using the resulting samples covering the surface of tin oxide (ITO) conductor glass. A quartz cell filled with 0.5 M Na_2_SO_4_ (pH = 6.8) electrolyte was used as the measuring system. The saturated photocurrent density of the 5-CdS/CdCO_3_@SDS (about 1.8 μA cm^−2^) was much higher than those of the CdS (about 1.0 μA cm^−2^) and CdCO_3_ (about 0.4 μA cm^−2^), which showed that the composite catalyst modified by surfactant can encourage the separation and transport of photogenerated electron-hole pairs [33,34]. In addition, the composite catalyst showed a repeatable and rapid photocurrent response under visible-light irradiation, which indicated that the sample had excellent photoelectrochemical stability. Electrochemical impedance spectroscopy (EIS) was shown to be an effective technique to study the charge-transfer resistance in the interface region [35] and a smaller arc on the EIS Nyquist diagram indicated a smaller charge-transfer resistance [36]. For the EIS measurements, the amplitude of the sinusoidal wave was 10 mV and the frequency ranged from 100 kHz to 0.1 kHz. As shown in Figure 5b, the relative arc radii of three catalysts were CdCO_3_ > CdS > 5-CdS/CdCO_3_@SDS, indicating that the interfacial charge-transfer rate was the highest in the 5-CdS/CdCO_3_@SDS. According to the above analysis and discussion, the 5-CdS/CdCO_3_@SDS possessed the best photoelectrochemical properties.

The prepared photocatalysts were used to determine the light-collecting ability and related *E*_g_ of the material by using UV-Vis DRS spectroscopy. The CdS exhibited a broad absorption scope, ranging from 220 to 820 nm (Figure 6a). The growth of the CdS on the surface of the CdCO_3_ had a negative effect on the visible absorbance. Therefore, the absorption intensities of the composites in the visible-light region were slightly weakened compared to those of the pure CdS. The composite photocatalysts still showed strong light-absorption capacity. Additionally, the *E*_g_ values of the photocatalysts were calculated according to the Kubelka–Munk equation (Figure 6b,c). The *E*_g_ widths of the CdS and CdCO_3_ were 2.24 eV and 4.84 eV, respectively.

### 3.4. Photocatalytic Mechanism

As a wide-band-gap insulator material, theoretically, CdCO_3_ cannot be excited under visible light, but some reports of the application of insulator materials in the field of photocatalysis in recent years have proven that the composite of the insulator and some photocatalyst materials can improve catalytic activity. Fan Dong and Hong Wang et al. believed that in the insulator/semiconductor heterojunction system, there was a shift in the energy band after the insulator and the semiconductor were recombined, which caused the excitation of the electron-hole pair on the insulator [37,38]. However, one of their study targets was the P–N junction structure; further investigation showed that the potential difference caused by the shift in the energy band can effectively excite and transfer electrons. The CdS and CdCO_3_ were both N-type materials; the band-shift theory was not applicable. Xueli Hu et al. studied an N-N-type (BaCO_3_/g-C_3_N_4_) composite material and proposed a new theory for the insulator synergistic catalysis [39]. They believed that BaCO_3_ was not directly involved in the photocatalytic reaction process, but in the process of material synthesis, some Ba^2+^ ions with smaller diameters can enter the interior of the unit cell through the wider plane in g-C_3_N_4_, resulting in lattice distortion, forming hole traps. Hole traps trapped electrons and encouraged the separation and transfer of electron-hole pairs inside g-C_3_N_4_, thereby enhancing the photocatalytic activity. Therefore, based upon previous research and the help of VESTA software, we boldly speculated as to the photocatalytic mechanism of CdS/CdCO_3_ composites [40]. During the preparation of composite photocatalysts, some Cd^2+^ with ion radii of 1.52 Å from CdCO_3_ can enter the internal unit cell of CdS (Figure 7a,b). The lattice distortion caused by ion entry forms many hole traps, which encourages the separation of electron-hole pairs on the surface of CdS (Figure 7c). At the same time, since the conduction band of CdS (−0.54 eV) was more negative than that of CdCO_3_ (−0.48 eV), some CdS electrons flow to the CdCO_3_. the recombination of electron-hole pairs inside CdS is inhibited and then encourages their transport. This is why the photocatalytic reduction ability of CdS can be greatly improved.

According to the above analysis, we propose a mechanism for the synergistic photocatalytic reduction of hexavalent chromium by CdS/CdCO_3_@SDS composite photocatalysts (Figure 8). On one hand, part of the Cd^2+^ ions from CdCO_3_ entered the CdS and optimized the separation and transport of its internal electron-hole pairs. On the other hand, under the action of surfactants, the morphology of the composite catalyst was changed. Its morphological structure was optimized to obtain a larger specific surface area, and the contact with the pollutant solution was more sufficient. H_2_O combined with holes to form oxygen (H_2_O/O_2_ 0.82 eV vs. NHE) [41]; O_2_ may also have originated from the solution itself. Oxygen molecules obtained electrons to generate superoxide radicals (O_2_/·O_2_^−^ −0.33 eV vs. NHE) [42]. ·O_2_^−^ reduced highly toxic Cr(VI) to low toxic Cr(III) and then themselves became O_2_. Of course, Cr(VI) can also be directly converted into Cr(III) by e^−^ (Cr(VI)/Cr(III) 1.33 eV vs. NHE) [43]. Hydroxyl groups do not directly participate in the reduction process of Cr(VI), but superoxide radicals react with hydrogen ions to generate H_2_O_2_ and O_2_. In an acidic environment, H_2_O_2_ is unstable and binds to holes to form ·OH. Therefore, the capture of hydroxyl groups encourages the consumption of a large number of superoxide radicals, which is not conducive to the reduction process of hexavalent chromium ions. Furthermore, the capture of hydroxyl groups encourages the consumption of a large number of superoxide radicals, which is not conducive to the reduction of hexavalent chromium ions.
(3)$$h_{\mathrm{vb}}^{+}+2H2O\;\to {\;O}_{2}{+4H}^{+}$$
(4)$$e_{\mathrm{cb}}^{-}{+O}_{2}\;\to {\;\cdot O}_{2\;}^{-}$$
(5)$${\mathrm{Cr}(\mathrm{VI})+\cdot O}_{2}^{-}\;\to {\;\mathrm{Cr}(\mathrm{III})+O}_{2}$$
(6)$${\mathrm{Cr}(\mathrm{VI})+e}_{\mathrm{cb}\;}^{-}\;\to \;\mathrm{Cr}(\mathrm{III})\;$$
(7)$$O_{2}^{-}{+\cdot O}_{2}^{-}{\;+\;2H}^{+}\to {\;2H}_{2}O_{2}{+O\;}_{2\;}$$
(8)$$H_{2}O_{2\;}{+\;h}_{\mathrm{vb}}^{+}\;\to \;\mathrm{OH}+\cdot \mathrm{OH}$$

## 4. Conclusions

In summary, series *X*-CdS/CdCO_3_ composites were constructed by chemical precipitation, which exhibited better photocatalytic efficiencies in Cr(VI) reduction than the single phase of CdS and CdCO_3_. This was mainly because the Cd^2+^ of CdCO_3_ entered the CdS crystal structure, produced lattice distortion and then formed hole traps and the electron flow of CdS itself. Especially after the modification of the surfactants, the reduction efficiency of the 5-CdS/CdCO_3_ was increased by about 10%, to 86.9%, while the reduction rates of 0.0675 were 15.57 times and 14.46 times those of the CdS and CdCO_3_, respectively. The BET, SEM and TEM analyses proved that the surfactant changed the morphology of the composite catalyst and increased the specific surface area. The UV-vis DRS proved that the composite catalyst had excellent visible-light-absorption intensity. The photocurrent and impedance experiments showed that the catalysts can significantly inhibit the recombination of photogenerated electron-hole pairs. The capture experiments determined that ·O_2_^−^ and ·OH were the main active substances in the photocatalytic reactions, while ·OH indirectly affected the reduction process of the Cr(VI). This work provides a feasible strategy for demonstrating the modification of photocatalysts by active agents, providing more prospects for CdCO_3_ in the field of water treatment.

## Figures and Tables

**Figure 1 nanomaterials-12-03923-f001:**
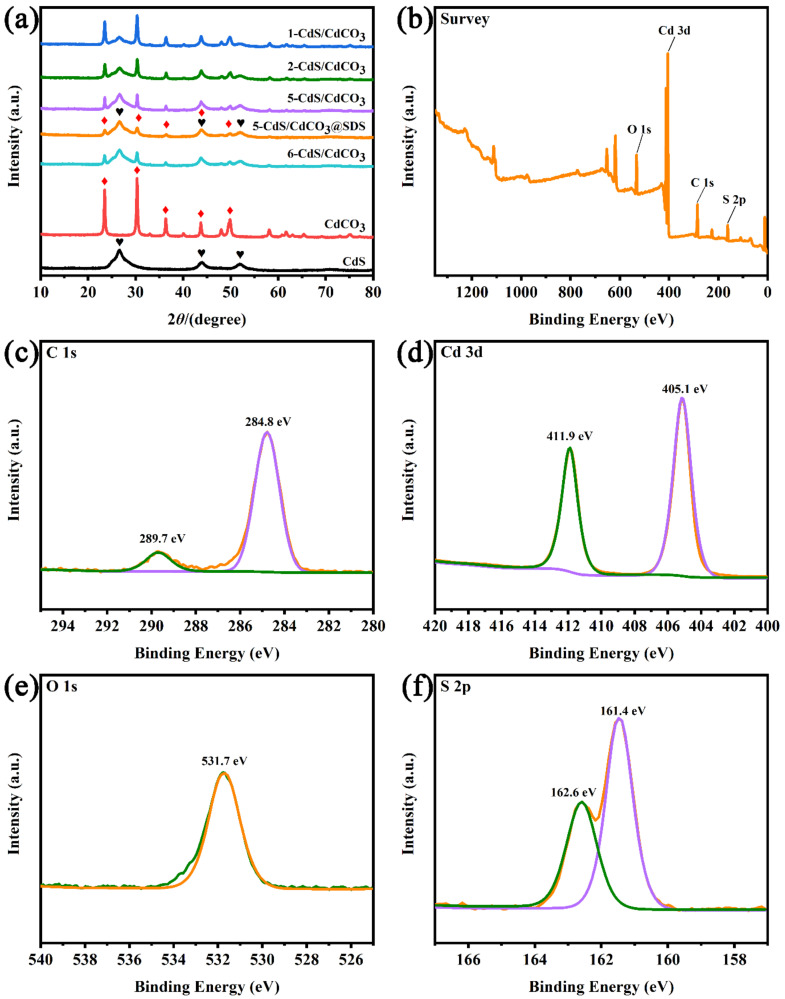
XRD patterns of CdS, CdCO_3_, *X*-CdS/CdCO_3_ and 5-CdS/CdCO_3_@SDS, red rhombus/black hearts correspond to CdCO_3_ and CdS respectively (**a**), XPS spectra of 5-CdS/CdCO_3_@SDS sample (survey spectrum) (**b**), C 1 s (**c**), Cd 3d (**d**), O 1 s (**e**) and S 2p (**f**).

**Figure 2 nanomaterials-12-03923-f002:**
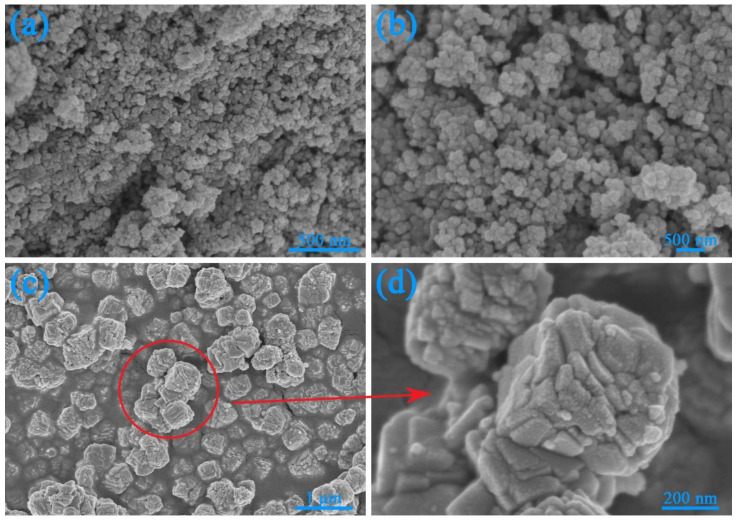
SEM images of CdS (**a**,**b**), CdCO_3_ (**c**,**d**), 5-CdS/CdCO_3_ (**e**,**f**) and 5-CdS/CdCO_3_@SDS (**g**,**h**).

**Figure 3 nanomaterials-12-03923-f003:**
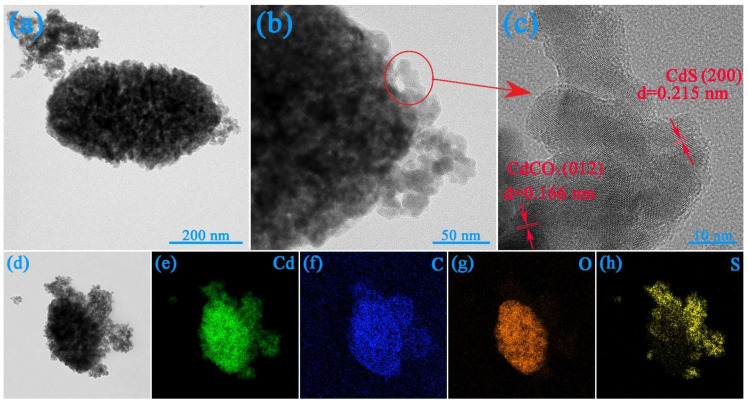
TEM of 5-CdS/CdCO_3_@SDS (**a**–**c**), HR-TEM and mapping elements (**d**–**h**).

**Figure 4 nanomaterials-12-03923-f004:**
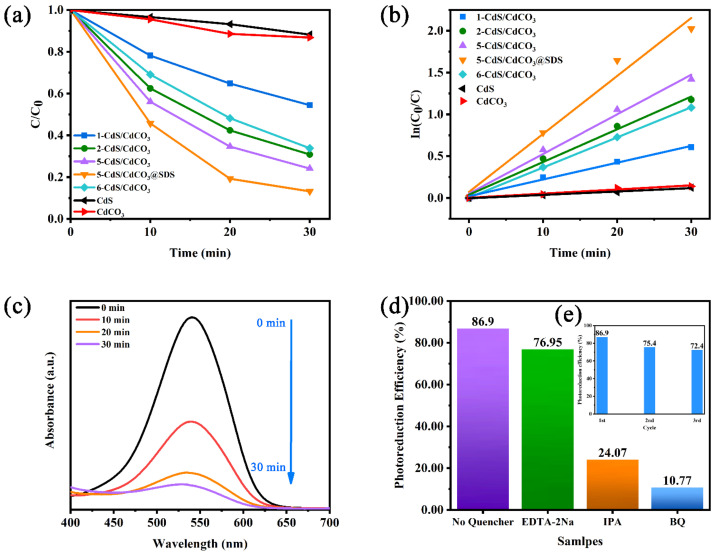
The change in Cr(VI) solution concentration with illumination on different samples (**a**), the corresponding kinetics of Cr(VI) degradation (**b**), the photocatalytic performance of 5-CdS/CdCO_3_@SDS (**c**), the effect of scavengers on the Cr(VI) degradation (**d**) and the cycling runs of -CdS/CdCO_3_@SDS (**e**).

**Figure 5 nanomaterials-12-03923-f005:**
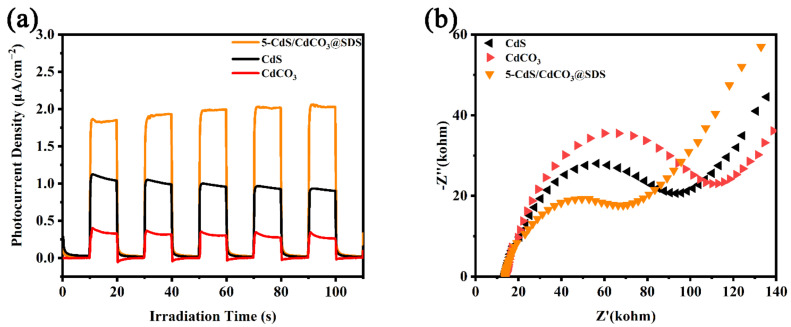
Photocurrent response (**a**) and EIS Nyquist plots (**b**).

**Figure 6 nanomaterials-12-03923-f006:**
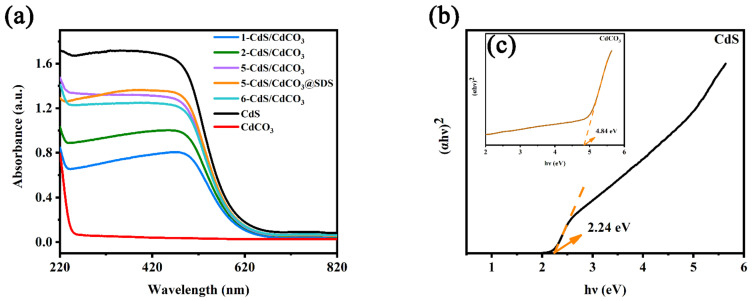
DRS spectra (**a**) and the band energy of as-synthesized photocatalysts (**b**,**c**).

**Figure 7 nanomaterials-12-03923-f007:**
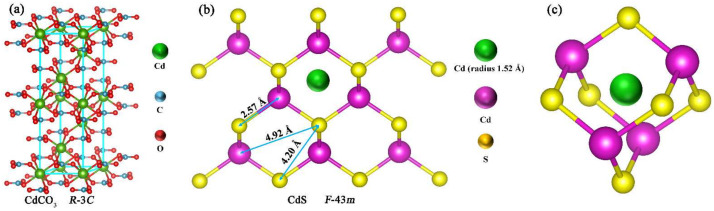
Crystal structures of CdCO_3_ (**a**) and CdS (**b**). Cd^2+^ (from CdCO_3_) enters the unit cell of CdS and causes lattice distortion (**c**).

**Figure 8 nanomaterials-12-03923-f008:**
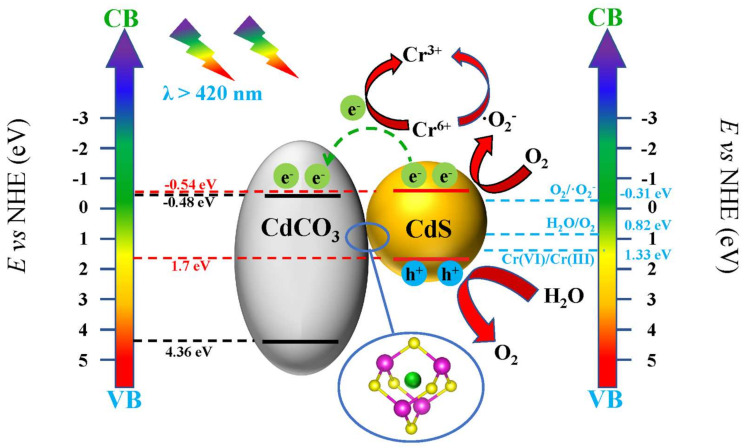
Schematic illustration of Cr(VI) photoreduction on 5−CdS/CdCO_3_@SDS under simulated sunlight irradiation.

**Table 1 nanomaterials-12-03923-t001:** Surface areas (*S*_BET_) of CdS, CdCO_3_, 5-CdS/CdCO_3_ and 5-CdS/CdCO_3_@SDS.

Sample	*S*_BET_/(m^2^g^−1^)
CdS	75
CdCO_3_	7
5-CdS/CdCO_3_	50
5-CdS/CdCO_3_@SDS	62

## Data Availability

The data that support the findings of this study are available from the corresponding author, (Y.Z.), upon reasonable request.

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
