# Peer review of "Surfactant-Modified CdS/CdCO3 Composite Photocatalyst Morphology Enhances Visible-Light-Driven Cr(VI) Reduction Performance"

_nanomaterials, 2022, doi:10.3390/nano12213923_

Round 1
Reviewer 1 Report
In the present article, the authors have reported the series of X-CdS/CdCO3 composites which were constructed by chemical precipitation, which exhibited better photocatalytic efficiencies of Cr (VI) reduction than the single phase of CdS and CdCO3. The results are presented well with justification. I recommend the work for publication; I have some points for further improvement.
1. In the experimental section there is no detail about the photoelectrochemical setup. Please write in details, how the measurement is done.
2. The reusability of the catalyst is very important factor. Did authors check the reusability limit of their catalyst.
3. There are small grammatical and typo errors, please improve it in the manuscript.
Author Response
Dear Reviewer
We sincerely thank the reviewer for thoroughly examining our manuscript and providing very helpful comments to guide our revision. We have tried our best to revise the manuscript according to your construction comments.
- We have supplemented and revised details on the photochemical setup in the new manuscript.
- In the newly uploaded manuscript, we uploaded the test results on the cycle life of the photocatalyst in the fourth figure.
- Thank you very much for discovering our grammatical and typed errors. We apologize for this grammatical problem and have corrected it based on our suggestions.
We would like to thank the referee again for taking the time to review our manuscript.
Reviewer 2 Report
Review of the Manuscript Nanomaterials-212606
1. English text should be improved, and Abstract, Introduction, Results and Conclusions sections should be rewritten.
Abstract, page 1, row 12-16
“Surfactant-modified catalyst morphology has been considered an effective method to improve photocatalytic performance. Here, the visible-light-driven composite photocatalyst was achieved by growing CdS nanoparticles on the cubic structure of CdCO3 and successfully changed the morphology of CdCO3 into elliptical spheres with surfactant modification.”
Should be replaced by
“Surfactant-modification of catalyst morphology has been considered as an effective method to improve photocatalytic performance. In this work the visible-light-driven composite photocatalyst was obtained by growing CdS nanoparticles in the cubic crystal structure of CdCO3, which after surfactant modification leads to formation ofCdCO3 elliptical spheres.”
2. Introduction, page 2, Row 29-31
“Environmental pollution caused by population expansion and industrial production is becoming more and more prominent, resulting in serious damage to the 30 natural water environment [1-5].”
Should be substituted with
“Environmental pollution caused by population expansion and industrial production is becoming more and more prominent, resulting in serious damage of the natural water environment [1-5].”
3. Please add text about the aim of this work in the end of Introduction section.
4. Experimental, Results, page 4, row 80-88
2.2 Preparation of CdS/CdCO3 nanocomposites
“Weighing 0.4567 g of CdCl2·2.5H2O, dissolve it in 50 mL of deionized water with 30 minutes stirring and call it solution A. Then weigh 0.2119 g Na2CO3 and dissolve it in 50 mL deionized water, stir for 30 minutes, then add 1.4446 g CdS, continue to stir for 30 minutes until fully mixed, this solution named solution B. Slowly invert solution A into solution, stir for 12 hours, filter it with a suction filter, and then place it in an oven at 60 °C for 12 hours and grind it.”
Should be replaced by
“0.4567 g of CdCl2·2.5H2O is dissolved in 50 mL of deionized water and the solution was subjected to 30 minutes stirring. The obtained solution is and denoted as solution A. 0.2119 g Na2CO3 is dissolved in 50 mL deionized water, stirred for 30 minutes, then 1.4446 g CdS, is added under continuous stirring until fully mixed - this solution is denoted as solution B. Solution A is slowly added into solution B, stirred for 12 hours, filtered with a suction filter, then dried in oven at 60 °C for 12 hours, and grinded. “
5. The BET surface area data has to be with accuracy of 1 m2/g. Please correct the data in Table 1.
6. page 15, row 304
“Conclusion” should be corrected as “Conclusions”

Author Response
Dear Reviewer
We sincerely thank the reviewer for thoroughly examining our manuscript and providing very helpful comments to guide our revision. We sincerely hope that this revised manuscript has addressed all your comments and suggestions. This manuscript has been revised extensively according to the reviewers' constructive suggestions. In addition, the expression of the manuscript has been improved with the help of a native English speaker.
We would like to thank the referee again for taking the time to review our manuscript.